# Reaction Kinetic Models of Antibiotic Heteroresistance

**DOI:** 10.3390/ijms20163965

**Published:** 2019-08-15

**Authors:** Antal Martinecz, Fabrizio Clarelli, Sören Abel, Pia Abel zur Wiesch

**Affiliations:** 1Department of Pharmacy, Faculty of Health Sciences, UiT—The Arctic University of Norway, 9037 Tromsø, Norway; 2Centre for Molecular Medicine Norway, P.O. Box 1137, Blindern, 0318 Oslo, Norway

**Keywords:** reaction kinetics, antibiotics, pharmacodynamics, Gillespie algorithm, antibiotic resistance, bacterial persistence, stochastic simulation

## Abstract

Bacterial heteroresistance (i.e., the co-existence of several subpopulations with different antibiotic susceptibilities) can delay the clearance of bacteria even with long antibiotic exposure. Some proposed mechanisms have been successfully described with mathematical models of drug-target binding where the mechanism’s downstream of drug-target binding are not explicitly modeled and subsumed in an empirical function, connecting target occupancy to antibiotic action. However, with current approaches it is difficult to model mechanisms that involve multi-step reactions that lead to bacterial killing. Here, we have a dual aim: first, to establish pharmacodynamic models that include multi-step reaction pathways, and second, to model heteroresistance and investigate which molecular heterogeneities can lead to delayed bacterial killing. We show that simulations based on Gillespie algorithms, which have been employed to model reaction kinetics for decades, can be useful tools to model antibiotic action via multi-step reactions. We highlight the strengths and weaknesses of current models and Gillespie simulations. Finally, we show that in our models, slight normally distributed variances in the rates of any event leading to bacterial death can (depending on parameter choices) lead to delayed bacterial killing (i.e., heteroresistance). This means that a slowly declining residual bacterial population due to heteroresistance is most likely the default scenario and should be taken into account when planning treatment length.

## 1. Introduction

When bacteria are exposed to antibiotics in vitro or in vivo, the elimination rate often changes dramatically over time. Typically, an initial phase of rapid decline is followed by a phase where bacterial killing is very slow or even absent. However, the bacteria that survive the rapid decline are not genetically different from those that were killed. When the bacteria are recultured from the surviving population and exposed to antibiotics at the same concentration again, they exhibit the same bi- or multiphasic killing as in the first experiment. This sets delayed bacterial killing apart from stable resistance mutations (antibiotic resistance). Just like antibiotic resistance, multiphasic bacterial killing has clinical implications: it is thought to complicate treatments, as it allows fractions of bacterial populations to survive extended exposure to antibiotics [1,2].

The characteristic slowdown of bacterial elimination after antibiotic exposure can be due to various mechanisms. According to a recent consensus statement [3], these mechanisms can be divided into either antibiotic persistence or heteroresistance. In this work we focus on the latter: in heteroresistance, the observation is that not all cells in a bacterial population are equally susceptible to antibiotics. It can be defined as the coexistence of multiple subpopulations of bacteria with varying levels of susceptibility to antibiotics. In these mixed populations, the more resistant subpopulations should have a significantly different susceptibility. In practice, this means that the so-called minimum inhibitory concentration (MIC; antibiotic concentration where the net growth of the population is zero) should show an at least eightfold increase when compared to the majority of the population [3,4,5]. There are multiple mechanisms that can contribute to this phenomenon, such as unstable resistance mutations [6]. In addition to mutations, it has also been shown that diversity among cells that result in slight variations in susceptibility to antibiotics can also lead to heteroresistance, including differences in the number of intracellular targets [7,8], differences in cell sizes and therefore diversity in the intracellular concentration of targets [9], or cell-to-cell differences in the number of efflux pumps that affect the intracellular antibiotic concentrations [10,11]. Currently, there is great interest surrounding heteroresistance; however, due to its by definition transient nature, it is difficult to investigate [1,4,5]. To find potential mechanisms that lead to heteroresistance and therefore guide experiments, mathematical modeling of antibiotic action can be helpful.

Indeed, models that include both intracellular drug-target binding and its effect on a bacterial population have been successfully employed to model heteroresistance [7]. Such drug-target binding models are becoming increasingly popular [12,13,14]. Modeling the reaction kinetics of drug-target binding can be challenging because it often involves low numbers of molecules; for example, the common antibiotic target gyrase has a 100 copies per cell [15,16]. In these cases, differential equations do not give accurate results, as they do not allow modeling discrete values instead of continuous values. This is a problem when the modeled process is sensitive to the changes in molecular numbers, for example, toxic byproducts and the model gives the result of half a molecule instead of 0 or 1. To circumvent this issue, approaches based on master equations can be employed [7]. For simplicity and the purpose of this paper, we call deterministic systems of differential equations “master equations” even if they are not necessarily linear. These deterministic equations model entire bacterial populations and classify bacteria into compartments according to how many target molecules are bound. However, these are often cumbersome to design and use, as they require a differential equation per “state” of the system, for example, one equation per possible number of bound targets.

Master equations have several advantages, among them the ability to incorporate bacterial growth and death with the subsequent changes in target molecule content and occupancy. As deterministic models, they do not need to be run many times to get representative averages. However, these current mathematical approaches have limitations in terms of the complexity of the molecular mechanisms that lead to bacterial death or suppress bacterial replication. When antibiotics bind to their intracellular targets, they interfere with essential functions of cells and thereby inhibit bacterial replication (bacteriostatic effect) or kill the cells (bactericidal effect). Both the bactericidal and bacteriostatic effects can be multi-step processes, for example the binding of antibiotics to their target can lead to the production of toxic byproducts that eventually kill the cells in, for example, kanamycin or norfloxacin [17]. In master equations-based approaches, the additional number of processes quickly inflates the required number of equations, as it would need one equation for each combination of molecule numbers, for example toxic byproducts and bound targets. The approaches used in [7] for example employ n+1 equations for n target molecules, with n ranging from 10^2^ to 10^5^. Modeling an additional step in the molecular cascade leading to bacterial death would then require 10^4^ to 10^10^ equations. Therefore, the current mathematical tools bar us from quickly testing and evaluating models of multi-step intracellular processes, thereby making it difficult to test theories on mechanisms that can lead to heteroresistance. As a result, while it is suspected that changes in the production of toxic byproducts or diversity in other parts in intracellular kinetics can lead to heteroresistant behavior, it is yet to be shown by modeling.

In this work, we use Gillespie simulations—stochastic simulations of the reaction kinetics in single bacterial cells [18,19]—to model multi-step intracellular processes and show that diversity in the rates in these processes at different levels can lead to heteroresistance. Thereby, we also demonstrate that Gillespie simulations are an effective tool in investigating population dynamics of bacteria, particularly when modeling multi-step processes. While Gillespie simulations have been an invaluable tool in computational biology, they are yet to be used in the context of modeling the response of bacterial populations to antibiotics.

## 2. Results

When bacteria are exposed to antibiotics, the elimination of bacteria can slow down dramatically over time. Figure 1a shows a schematic of a so-called time-kill curve (measured bacterial numbers over time after antibiotic exposure—and thereby gives an example of how multiple subpopulations can result in a slowdown of elimination. If this phenomenon is caused by heteroresistance, the underlying mechanism is a coexistence of multiple subpopulations with varying degrees of susceptibility to antibiotics. The susceptibility to antibiotics can be captured with pharmacodynamic curves that describe the net population growth (replication minus death, or the slope of the time-kill curve) at a given antibiotic concentration. The antibiotic concentration at zero net growth is called the minimum inhibitory concentration (MIC). Figure 1b illustrates how multiphasic killing of bacterial populations can lead to different pharmacodynamic curves, one for quickly and one for slowly declining bacteria. Here, we model single- (Figure 1c) and multi-step processes (Figure 1d) that result in the elimination of bacteria and obtain pharmacodynamic curves. We assess whether hypothetical variations in the parameters of the multistep processes involved in bacterial drug-susceptibility can result in bi- or multiphasic time-kill curves. We employ Gillespie simulations, demonstrating that they can be an effective tool in modeling antibiotic action in bacteria.

First, we show the relationship among Gillespie simulations and the already established methods—master equations and simple reaction kinetics. This relationship is well studied in the literature modeling other processes [19,20], and our goal here is to provide a starting point for comparing the models that are built upon them. Figure 2 illustrates that all methods model the same process, therefore yielding similar results. Simple reaction kinetics describe the mean (Figure 2a,b), and master equations describe the probability of observing a given state of the system (given number of bound targets) (Figure 2c,d). Gillespie simulations are stochastic simulations that describe one possible time-course of the observed number of bound targets over time (Figure 2e,f). When repeated sufficiently often, the average of the Gillespie simulations should yield the same result as the other two approaches.

We aimed to illustrate that modeling based on Gillespie algorithms predicts the same time–kill curves resulting from reaction kinetics as the already established heteroresistance model based on master equations [7]. To this end, we chose a setting where bacterial growth is negligible over the observed time period (easily modeled with classical Gillespie algorithms) and where bacterial death is caused by a single reaction step (easily modeled with master equations). We furthermore assume that when bacteria die, target degradation is negligible in the period we observe. This may mean either that the entire cellular structure remains intact for a while or that targets that are released by bacterial lysis into the medium are not degraded quickly. For the sake of clarity, we simulated the binding kinetics of a fictive situation with only few targets. The time course of target occupancy is shown in Figure 3a. The predictions of target occupancy are the same for a deterministic master equation model and the Gillespie algorithm, in line with the large body of literature showing that Gillespie algorithms are useful tools to describe reaction kinetics [19,23,24]. Based on these simulations of target occupancy, we predict time–kill curves of a bacterial population with multiple subpopulations having the same number of targets, but the thresholds for killing follow a normal distribution, that is, varying susceptibility or heteroresistance. In Figure 3, we demonstrate that both master equations and the Gillespie algorithm give essentially the same result.

We thereby established that both methods, as expected, are equivalent under the condition that bacterial replication can be neglected. Next, we set out to investigate how these dynamics are changed by bacterial replication: the production of new (unbound) target molecules and the inheritance of bound target molecules from the mother cell.

To illustrate under which conditions bacterial replication can be neglected, we now compare target occupancy predicted with master equation-based simulations for fast (doubling time 20 min, e.g., *Escherichia coli*) and slow (doubling time 24 h, e.g., *Mycobacterium tuberculosis*) bacterial replication rates. Figure 4 shows the differences between simulations where we kept the difference of bacterial replication and elimination constant (“same” net growth), but we changed the ratio of the two (i.e., high replication and elimination at the same time, or low bacterial elimination and low replication). While the assumption that bacterial replication does not affect target occupancy has been made elsewhere [13,26], we find that neglecting replication gives substantially different results for a doubling time of 20 min. However, we also find that there are relatively minor differences when replication only occurs every 24 h.

Finally, to show the flexibility of the approach and to investigate potential sources of heteroresistance, we modelled four different scenarios that lead to bacterial death (Table 1 in Materials and Methods shows the events and their rates modeled in the Gillespie algorithm):A high proportion of bound versus unbound target is toxic (i.e., “relative toxicity”). In this case, we assume a bacterium is dead once the ratio of bound target and total target molecules exceeds a threshold. Here, we introduce heterogeneity by assuming that the number of target molecules is normally distributed around its mean.The target has an “essential” function (i.e., the cell dies when less than a certain amount of target is free). In this case, we assume a bacterium is dead once the number of free target falls below a threshold. Here, we introduce heterogeneity by assuming that the threshold for killing is normally distributed around its mean.The antibiotic-target complex is “primary toxic” (i.e., the cell is immediately damaged by bound targets). In this case, we assume a bacterium is dead once the number of bound targets exceeds a threshold. Here, we introduce heterogeneity by assuming that the number of target molecules is normally distributed around its mean.The antibiotic-target complex is “secondary toxic” (i.e., triggers the accumulation of toxic metabolites that ultimately kill the cell). In this case, we assume a bacterium is dead once the number of toxic metabolites exceeds a threshold. Here, we introduce heterogeneity by assuming that the rate of toxic metabolite production is normally distributed around its mean.

As demonstrated previously for variances in target molecule content, normally distributed population variances in the binding kinetics of any of the steps leading to bacterial killing leads to multiphasic kill curves (or heteroresistance) (Figure 5). In our case, this occurred when the variation in the population was above 4−10% (Figure A2).

## 3. Discussion

The aim of this paper is to extend the toolbox for modeling the intracellular processes that kill bacteria and to illustrate how these tools can be used to describe bacterial heteroresistance. Traditionally, antibiotic efficacy was mainly described by a single value, the minimal inhibitory concentration (MIC, see points in Figure 1B). While there is often a correlation between treatment success and MIC [27,28], there is limited predictive power, since not all patients infected with bacteria that are classified as susceptible by their MIC are successfully treated with antibiotics [6]. More importantly, up to two thirds of patients infected with bacteria that are classified as resistant to the prescribed antibiotic based on MIC measurements are successfully treated (despite the infection per se being serious) [29]. This has led to calls for more sophisticated approaches where the entire dose-response curve (entire curve in Figure 1B) is taken into account, which is described by pharmacodynamic models [30]. Currently used pharmacodynamic models (E_max_/Hill-models) originated in 1910 [31,32] and make a large number of simplifying assumptions. Therefore, novel models including drug-target binding have been developed to overcome the limitations of traditional models.

Previously, established deterministic antibiotic-target binding models classify bacterial populations in compartments based on drug occupancy. Here, we use the Gillespie stochastic simulation algorithm that focuses on the reaction kinetics in single cell and model bacterial populations by running simulations as many times as there are cells in the population. This means that a bacterial population of e.g., 1000 cells is modeled by running 1000 individual simulations. The Gillespie algorithm is a well-established computational method in computational biology [24], chemistry [19], and epidemiology [33]. However, it has not been used to model the effects of antibiotics on bacterial populations. Gillespie simulations fill the niche of modeling multi-step intracellular processes, as the use of current modeling approaches make this cumbersome.

We have modeled bacterial death as an event that is triggered by spending a specific amount of time spent above a threshold of bound targets or toxic byproducts. This approximation is only valid for one generation of bacteria because it does not allow taking into account the fact that the bound targets of the mother cells are distributed to the daughter cells. This means that for the daughter cells, the simulations should not start at 0 bound targets. To accurately model this, it would be necessary to stop the simulation after the cell is supposed to replicate, dividing the bound targets randomly between the daughter cells, increase the total number of target to reflect target synthesis during growth, and then start new simulations with the daughter cells. This would greatly increase the complexity of the model and shows the central difference in the biological population aspects, that is, bacterial replication. In approaches based on master equations, this can be easily done for each compartment of bound target molecules with well-established biological population equations (e.g., logistic or exponential growth). Therefore, to demonstrate the how this approximation on growth changes bacterial dynamics, we modelled it with an established master equations-based approach [7].

In Figure 4, we presented four cases where we modeled how fast and how slow bacterial growth combined with different assumptions about target stability after cell death affects the dynamics of the system. Our aim here is to illustrate how and when to use different modeling approaches to describe antibiotic action on a molecular scale. We highlight how a major assumption, that antibiotic target occupancy is independent of bacterial replication, can fundamentally change model predictions. While this assumption has been made elsewhere [13,26], our results indicate that Gillespie simulations neglecting bacterial replication are not suited for modeling organisms with fast growth such as *E. coli*.

However, Gillespie algorithms seem to yield similar results as master equations for slow replication. When we assume that bacterial death results in immediate target degradation, the total amount of target binding is slightly underestimated. Conversely, when we assume the other extreme, that targets are stable after bacterial death, the total amount of target binding is overestimated. Therefore, if realistic target degradation rates were available to inform models with replication, they would probably result in an even better concordance between Gillespie simulations and master equations. However, the results presented here are strongly dependent on the parameter setting we chose. Before making a decision to neglect replication (and therefore making modeling complex reactions possible), it is advisable to compare one-step simulations parametrized with corresponding binding rates (e.g., the first step, a composite binding rate, or the binding rates of the rate-limiting step) with and without the replication rates of the bacterial organism to be studied. Figure A1 gives a more comprehensive overview of the dynamics when we change assumptions about growth and death. In summary, neglecting replication warrants careful checking for the specific bacteria–drug pair and is most applicable to slow bacterial replication rates as for example found in *M. tuberculosis*.

The intended uses of this approach are in vitro studies that investigate antibiotic effects on bacteria. For instance, the expression of genes involved in the bacterial response to antibiotics may vary in a bacterial population thereby leading to heteroresistance [6]. These measured heterogeneities can be used to parametrize the presented multistep models in order to predict how these may affect time–kill curves. They can be used to assess whether the measured heterogeneities are sufficient to explain response to antibiotics (as there could be a multitude of processes contributing to changes in time–kill curves). Additionally, in vivo gene expression levels may be measured [34], but it might be difficult to obtain time–kill curves. In these cases, our approach might help to better understand bacterial responses. Our and other drug-target binding models can also easily replace traditional pharmacodynamic E_max_- or Hill functions when modeling antibiotic action in patients. Moreover, they can be easily incorporated in mixed effect pharmacokinetic/pharmacodynamic (PK/PD) models where the drug concentration is supplied by a PK model, and drug-target binding models are used to describe fixed pharmacodynamic effects with additional stochastic effects [35].

Using our Gillespie-based approach, we have run simulations for both single-step processes as well as multi-step processes and demonstrated (Figure 5) that normally distributed population variances in the binding kinetics of any of the steps involving bacterial killing can lead to multiphasic kill curves (or heteroresistance). This means that heteroresistant behavior can emerge in response to minor changes in the processes leading to bacterial death. As we have shown previously in [7,8], this behavior can be expected at concentrations where the pharmacodynamic curves for the majority and minority subpopulations (for example, Figure 1b) are further apart from each other. This mainly happens at the steeper parts of the pharmacodynamic curves (i.e., around inflection point), where a shift in the pharmacodynamic curves caused by the changes in binding kinetics can create a large enough difference thereby producing multiphasic time–kill curves.

We would therefore caution not to interpret the criterion of a substantially different MIC in heteroresistant subpopulations too strictly, in concordance with [6,36]. An eightfold change in MIC as an experimental criterion is sensible because of the low accuracy of testing MICs (fourfold changes are regarded as the same result [36]). However, our models show that minor changes in binding kinetics, which do not necessarily lead to an 8-fold change in MIC, can lead to multiphasic time-kill curves (i.e., heteroresistance). In addition, our work highlights that one should expect multiphasic kill curves in almost all settings even in the absence of specific bacterial subpopulations, because it is extremely unlikely that the content of all molecules involved in bacterial killing is exactly the same in all cells of a population. Therefore, all antibiotic treatment should take residual, slowly killed bacterial populations into account.

## 4. Materials and Methods

### 4.1. Simple Reaction Kinetics (Excluding Bacterial Replication and Death)

For modelling simple reaction kinetics between antibiotic molecules and their target molecules, we used the following differential equations:(1)$$\frac{dAT\left(t\right)}{dt}=A\cdot T\left(t\right){\hat{k}}_{f}-AT\left(t\right)k_{r}$$
(2)$$\frac{dT\left(t\right)}{dt}=-A\cdot T\left(t\right){\hat{k}}_{f}+AT\left(t\right)k_{r},$$
where:${\hat{k}}_{f}=\frac{k_{f}}{V_{i}\cdot n_{A}}$ is the adjusted forward reaction rate to accommodate working with the number molecules inside the cells instead of concentrations. Here, $k_{f}$ is the forward reaction rate, $V_{i}=10^{-15}\;l$ is the average cell volume, and $n_{A}=6\cdot 10^{23}$ is the Avogadro number.$k_{r}$ is the reverse reaction rate.A is the number of antibiotic molecules inside the cell.T(t) is the number of free target molecules inside the cell.AT(t) is the number of bound target molecules inside the cell.

For all simulations we used the binding parameters of ciprofloxacin: the parameters used are $K_{D}=\frac{k_{r}}{k_{f}}=10^{-5}$ M (ciprofloxacin [21,22]), number of targets *n* = 100 (gyrase, [15,16])).

We only used the simple reaction kinetics for illustrating the connection among the different ways to model chemical reaction kinetics, therefore this model does not include bacterial replication or death.

### 4.2. Master Equations (Including Bacterial Replication and Death)

An approach based on systems of ordinary differential equations fully described in [7] combines a modeling the chemical reaction kinetics of antibiotics binding to their targets, and bacterial population dynamics (i.e., bacterial growth and death). Here, each equation describes bacteria with a given number of bound targets. However, in addition to the reaction kinetics model providing the connection between the equations, upon replication the mother cells distribute their bound targets among the daughter cells randomly.
(3)$$\frac{dB_{x}}{dt}={\hat{k}}_{f}\left(n-x+1\right)AB_{x-1}-k_{r}xB_{x}-{\hat{k}}_{f}\left(n-x\right)AB_{x}+k_{r}\left(x+1\right)B_{x+1}+\rho_{x}\;-r_{x}B_{x}\frac{K-\sum_{j=0}^{n}B_{j}}{K}-d_{x}B_{x},\;\mathrm{for}\;x\in \left[2:n-1\right]$$
$$\frac{dB_{0}}{dt}=-{\hat{k}}_{f}nAB_{0}+k_{r}B_{1}+\rho_{0}-r_{0}B_{0}\frac{K-\sum_{j=0}^{n}B_{j}}{K}-d_{0}B_{0},\;\;\;\;\;\;\;\;\;\;\;\;\;\;for\;x=0\;\frac{dB_{n}}{dt}={\hat{k}}_{f}AB_{n-1}-k_{r}nB_{n}+\rho_{n}-r_{n}B_{n}\frac{K-\sum_{j=0}^{n}B_{j}}{K}-d_{n}B_{n},\;\;\;\;\;\;\;\;\;\;\;\;\;for\;x=n$$
where:x is the number of bound targets,n is the number of targets,$B_{x}$ is the number of bacteria with x bound targets,$d_{x}$ is the elimination rate of bacteria at x bound targets,$\rho_{x}=2\sum_{i=x}^{n}f_{i,x}r_{i}B_{i}\frac{K-\sum_{j=0}^{n}B_{j}}{K}$—describes the inheritance of bound targets from the mother cells (“inflow”) using the hypergeometric distribution $f_{i,x}$ for the random distribution of targets.$r_{x}$ is the replication rate of cells at x bound targets.

These systems of differential equations are not necessarily classical master equations because they are not linear. However, when assuming that bacterial replication is exponential and not logistic (i.e., there is no maximal bacterial density) and when assuming that the extracellular antibiotic concentration is constant, these systems become linear and therefore classical master equations. Similarly to [7,8], we also modeled heteroresistance with these equations, here we modeled multiple subpopulations with varying numbers of targets, threshold for elimination, or rate. Here we describe the frequency of subpopulations as a normal distribution around the mean. We varied the standard deviation to model different levels of heterogeneity.

### 4.3. Gillespie Simulations (Excluding Bacterial Replication And Including Bacterial Death)

The Gillespie stochastic simulation algorithm first published in 1977 is a general algorithm to simulate stochastic processes originally developed to capture the binding kinetics [18]. Starting from the initial conditions, the simulation is computed by repeating the following three steps. First, determining the rates of the next possible reactions based on the number of molecules and reaction rates ($R_{i}$) at the given time-point. Second, drawing two random numbers, one from an exponential distribution with a mean of ${\left(\sum R_{i}\right)}^{-1}$ to determine the time until the next time-step. In addition, we chose which reaction is going to happen, where reaction “x” taking place has the probability of $\frac{R_{x}}{\sum R_{i}}$. The third step involves updating the numbers of molecules and time accordingly to step two [19].

In this work, we use this algorithm to simulate chemical reaction kinetics within cells, modeling both systems with a single or two steps (Table 1).

### 4.4. Implementation

Codes for Figure 3, Figure 5, and S2 were implemented in statistical software package R (version 3.4.4, The R Foundation for Statistical Computing, Vienna, Austria). Codes for Figure 2 were implemented in Mathematica (Version 12.0, Wolfram Research, Inc., Champaign, IL, USA). Codes for Figure 4 and S1 were implemented in MATLAB (2019a, The MathWorks, Natick, MA, USA). All codes are available on www.abel-zur-wiesch-lab.com.

## Figures and Tables

**Figure 1 ijms-20-03965-f001:**
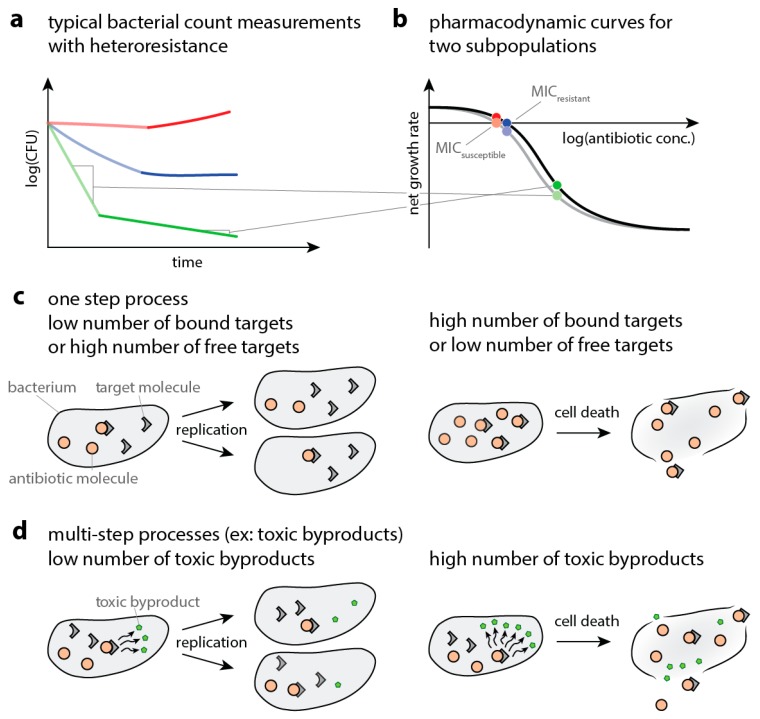
Schematic explanation of observed data and underlying processes. (**a**) Schematic drawing of a time–kill curve, where bacterial numbers are followed over time after exposure to different antibiotic concentrations (increasing from red to blue to green). The initial fast decline is depicted in pale, the subsequent slow decline in dark colors. (**b**) Schematic of a pharmacodynamic curve showing how increasing antibiotic concentrations (different colors) affect bacteria. The x-axis shows logarithm of the antibiotic concentrations. The y-axis shows the resulting net growth rate, that is., bacterial replication minus death, as measured by the slope of bacterial decline. The gray line depicts the quickly declining population (more susceptible), the black line represents the population declining more slowly. The grey lines between (**a**) and (**b**) illustrate how to translate the slope of time–kill curves to the pharmacodynamic curves. The pale and dark points illustrate the slopes of the time–kill curves in the corresponding colors in (**a**). (**c**) Schematic representation of the events leading to bacterial death if the drug-target complex itself is toxic. Bacteria can still replicate when few target molecules are bound by the antibiotic (left panel) but die when a high number of target molecules that exceed the killing threshold are bound (right panel). (**d**) Schematic representation of the events leading to bacterial death if the drug-target complex causes the accumulation of a toxic metabolite. Bacteria can still replicate when only a few metabolite molecules are present and die when too much metabolite has accumulated.

**Figure 2 ijms-20-03965-f002:**
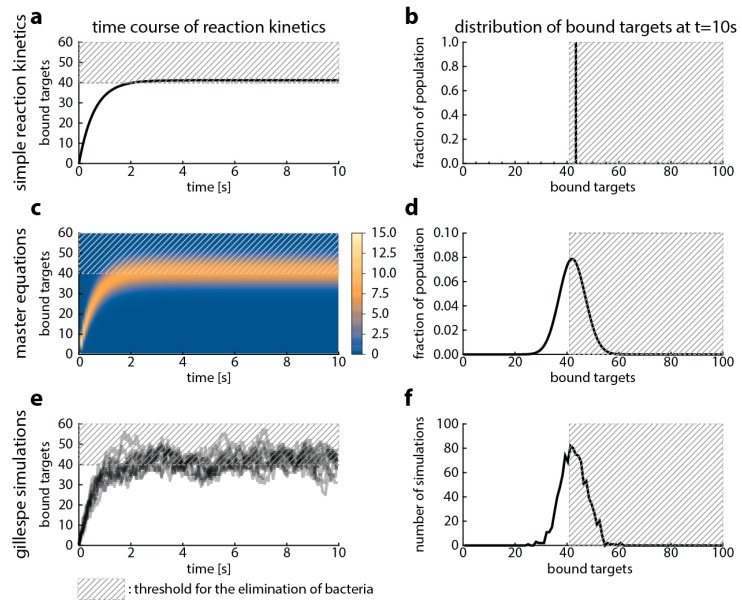
Comparison of modeling approaches that follow the reaction kinetics of antibiotic target-binding. In this figure, we compare three approaches: simple reaction kinetics (**a**,**b**), master equations (**c**,**d**) and Gillespie simulations (**e**,**f**). We used the following parameters: $k_{f}=10^{5}{\;M}^{-1}s^{-1}$ and $k_{r}=1{\;s}^{-1}$ (based on $K_{D}=\frac{k_{r}}{k_{f}}=10^{-5}\;M$ for ciprofloxacin [21,22]), number of targets: 100 molecules (gyrase, [15,16]), and the antibiotic concentration is 116 μM. The left column shows the time course of antibiotic target-binding. The x-axis shows the time, the y-axis shows the percentage bound target and the dashed grey area illustrates an arbitrary threshold of bound target when we presume bacterial killing would occur. The legend for the heat map in (**c**) is given next to the figure and shows the percentage of the bacterial population in a given state. The right column shows the distribution of target occupancies (x-axis) in the upper panel for the corresponding modeling approach well after steady state has been reached (plateau in left column, at 10 s).

**Figure 3 ijms-20-03965-f003:**
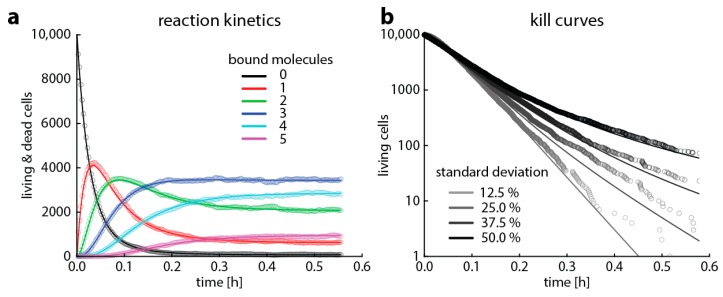
Comparison of the output of a Gillespie algorithm and simulations based on master equations. This is a simplified case for clarity, where a cell only contains five target molecules and dies on average within one second when more than two target molecules are bound. The binding rate k_f_ is 1.2 × 10^6^ M^−1^s^−1^, the unbinding rate k_r_ is 1.2 × 10^−3^ (for rifampicin, [25]), and the antibiotic concentration is 1.7 nM (10^15^ molecules/liter). (**a**) Shows the predicted average target occupancy: the number of bacteria with 0 (black), 1 (red), 2 (green), 3 (blue), 4 (turquois), and 5 (purple) bound targets for 10,000 Gillespie simulations (dots) or the master equation approach (lines). (**b**) Shows the predicted time–kill curves when modeling 10,000 bacterial cells with Gillespie simulations (dots) or the master equation approach (lines) for bacterial populations where the threshold for killing follows a normal distribution with 12.5%, 25%, 37.5%, or 50% standard deviation.

**Figure 4 ijms-20-03965-f004:**
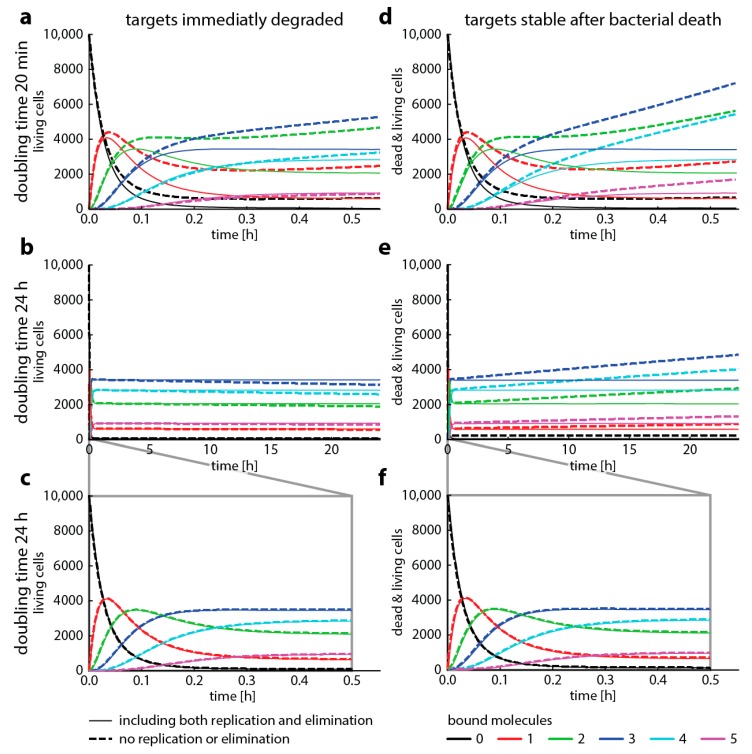
Bacterial replication changes target occupancy. This graph shows how target occupancy with replication and death (dashed lines) changes compared to target occupancy without replication or death (solid lines, compare to Figure 2) for two different cases. Cases, where the drug targets are immediately degraded after bacterial death (left column) or remain stable after bacterial death (right column). All parameters are the same as in Figure 3, with added replication and adjusted death rates. We assume that replication immediately ceases when more than two target molecules are bound and that cell die immediately when there are at least 4 bound target molecules. The doubling time ($\frac{ln\left(2\right)}{replication\;rate})$ is either 20 min (top panel) or 24 h (mid and bottom panels). The bottom panel highlights the first 30 min of the mid panel time course. The death rate is always 2 *x* the replication rate. (**a**–**c**) show the number of living bacteria only, (**d**–**f**) show both living and dead cells.

**Figure 5 ijms-20-03965-f005:**
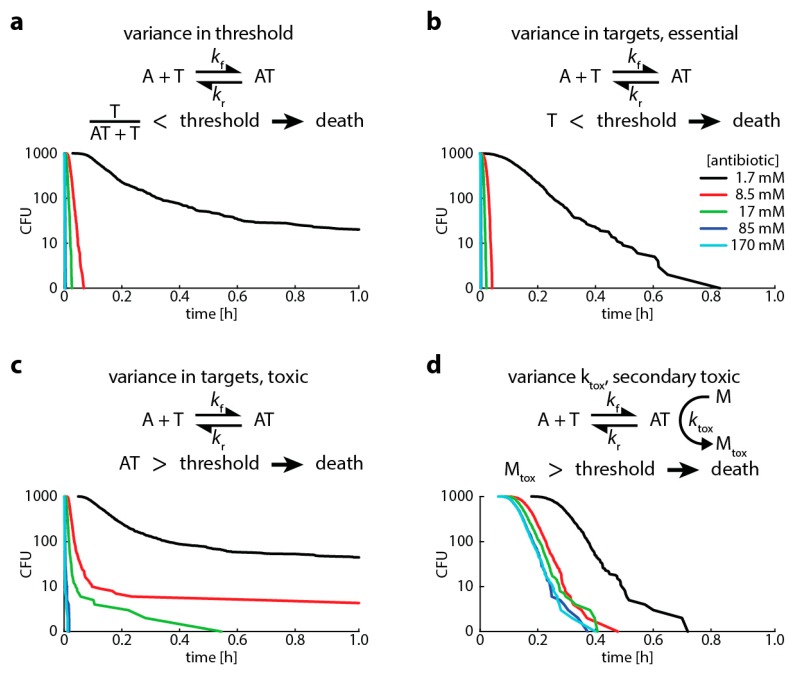
Variances in any step that causes bacterial killing: binding kinetics or killing threshold leads to multiphasic kill curves. This graph shows Gillespie simulations of four different scenarios of bacterial killing (see explanation in text). In all simulations, we assume (i) 100 target molecules, (ii) a binding rate k_f_ of 1.2 × 10^6^ M ^−1^s^−1^, (iii) an unbinding rate k_r_ of 1.2 × 10^−3^ (for rifampicin, [25]), and (iv) an intracellular volume of 10^−15^. We vary the antibiotic concentration from 1.7 (black), 8.5 (red), 17 (green), 85 (blue) to 170 (turquoise) mM. (**a**) “Relative toxicity” (point 1 in main text above), threshold for killing 50%, threshold normally distributed with 20% standard deviation (SD). (**b**) “Essential target function” (point 2 in main text above), threshold for killing 50 free target molecules, target number normally distributed with 20% SD. (**c**) “Primary toxic” (point 3 in main text above), threshold for killing 50 bound target molecules, target number normally distributed with 20% SD. (**d**) “Secondary toxic” (point 4 in main text above), threshold for killing 500 accumulated toxic metabolites, rate of metabolite production k_tox_ is 1.2 × 10^6^ M^−1^s^−1^, k_tox_ normally distributed with 20% SD.

**Table 1 ijms-20-03965-t001:** Event table and parameter descriptions for Gillespie simulations.

Model	Reactions	Description	Rate
Single step model	A+T→AT	Forward reaction	${\hat{k}}_{f}\;AT$
	AT→A+T	Backward reaction	$k_{r}\;AT$
Multi–step model	A+T→AT	Forward reaction	${\hat{k}}_{f}\;AT$
	AT→A+T	Backward reaction	$k_{r}\;AT$
	$AT\to AT+M_{tox}$	Toxic metabolite production	$k_{tox}\;AT$

Using the results from each simulation, we implemented the elimination of bacteria taking place after the number of bound targets (single-step case) or toxic metabolites (multi-step case) cross a threshold. In a simplified case, the first time-point where the simulations cross this threshold, we consider the cell to be dead. However, in single-step processes, (computationally) there is no upper limit to the elimination rates as it is defined only by the speed of the reaction. Therefore, it is best implemented after as “time above the threshold” instead: after a cell spends more than a given amount of time above the threshold, we consider the cell dead.

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
