# Peer review of "Reaction Kinetic Models of Antibiotic Heteroresistance"

_ijms, 2019, doi:10.3390/ijms20163965_

Round 1
Reviewer 1 Report
In their manuscript “ Reaction kinetic models of antibiotic heteroresistance”, Martinecz et al use Gillespie simulations to model multi-step intracellular processes showing how variability in the different processes involved in bacterial killing can lead to heteroresistance and compare this approach to a previous one where the master equation was used.
The manuscript is well structured and discussed. I only have some comments/concerns that I would like the authors to discuss
1. How is bacterial replication or death implemented in the model and what are the corresponding constants used? In eq 1-2 only the dynamics of AT and T are given.
2. Could the authors also discuss how this mathematical model could be used with real experimental data to estimate the parameters controlling these pharmacodynamic processes and be linked to PK models to guide dose selection?
3. What are advantage of this approach over other well-established methods such as non-linear mixed effect modelling – where the implementation of replication or multi-step processes is not limiting- to guide dosing optimization?
Author Response
In their manuscript “ Reaction kinetic models of antibiotic heteroresistance”, Martinecz et al use Gillespie simulations to model multi-step intracellular processes showing how variability in the different processes involved in bacterial killing can lead to heteroresistance and compare this approach to a previous one where the master equation was used.
The manuscript is well structured and discussed. I only have some comments/concerns that I would like the authors to discuss
How is bacterial replication or death implemented in the model and what are the corresponding constants used? In eq 1-2 only the dynamics of AT and T are given.Thank you for your comments. We are using 3 different modelling approaches, which we call “simple reaction kinetics” (section 4.1), ”Master equations” (section 4.2) and “Gillespie simulations” (section 4.3). We apologize for the confusion- only 1 of these models, Master equations, incorporate bacterial replication. The equations the referee refers to belong to the “simple reaction kinetic” model and therefore do not include replication. This model was only used to show the relationship among the different ways to model chemical reaction kinetics. To clarify this, we have added a sentence to the materials and methods section on page 11 lines 1-2, as well as clarified it in the subheadings of the methods section. In the Master equations approach, bacterial replication is described in equation (3) with the parameter rx (line 18, page 11). The parameter values are given in the caption of each figure.
Could the authors also discuss how this mathematical model could be used with real experimental data to estimate the parameters controlling these pharmacodynamic processes and be linked to PK models to guide dose selection?We have now added a paragraph on page 9 line 47 to page 10 line 2. It is possible to parametrize our model with experimental data, specifically when changes in the number of molecules in bacteria can be measured. For example, changes in gene expression in different environments or unstable gene amplifications may lead to diverging molecular content. The linking to PK models is now discussed in the subsequent sentences (page 10 lines 3-7) along with the discussion how our approach can be incorporated into other modelling approaches (see point below).
What are advantage of this approach over other well-established methods such as non-linear mixed effect modelling – where the implementation of replication or multi-step processes is not limiting- to guide dosing optimization?In most PK/PD models that are currently used (also mixed effects models, that include a random component), the pharmacodynamic component is described with Hill or Emax functions that originated in 1910 and only accurately reflect reaction kinetics under several simplifying assumptions (discussed on page 8, line 25 – page 9 line 2). Our approach can be useful in obtaining more realistic PD curves that can be used in PK/PD methods, offering an alternative to PD models based on Hill functions. Therefore, our approach is intended to complement the already established PK/PD methods. We have now clarified this together with the response to point 2 (page 9 lines 47 to page 10 line 7).
Reviewer 2 Report
1. This study of heteroresistance and the comparison of using master equation and Gillespie algorithm is novel.
2. It is recommended that the authors would identify which figure they are referring to in the discussion part to avoid confusion.
3. To improve the readability, the authors should breakdown long sentences (for example page 7, line 3 to 7)
4. The authors are recommended to move the content in the Result part (page 6 line 7 to 15) to Discussion.
5. A more concrete explanation for page 8 line 19-20 stated clearly in Discussion instead of just a reference is recommended.
6. The Discussion part seemed to have similar content as the Introduction and do not discuss the results in details. However, the Results part was too lengthy for discussion was made in the Result part. It would be better for the authors to restructure the content in the Results and Discussion part.
7. The authors should provide a concrete conclusion of the use of this model in practice.
8. After restructuring, this manuscript can be accepted and would serve as a valuable literature for the pharmacodynamics studies.
9. Although the MIC method may logically have weaknesses, it is very easy and comparable to clinical data. A concrete example or conclusion is needed to see how the model presented in this paper can be used in clinical practice.
Author Response
This study of heteroresistance and the comparison of using master equation and Gillespie algorithm is novel. It is recommended that the authors would identify which figure they are referring to in the discussion part to avoid confusion.Thank you for the comment, now we have amended this throughout the discussion.
To improve the readability, the authors should breakdown long sentences (for example page 7, line 3 to 7)Now we have broken up this sentence (now page 6 lines 15-19) as well as several others throughout the manuscript.
The authors are recommended to move the content in the Result part (page 6 line 7 to 15) to Discussion.Now this content has been moved to the discussion (page 9 lines 33-46).
A more concrete explanation for page 8 line 19-20 stated clearly in Discussion instead of just a reference is recommended.We expect heteroresistant behaviour when differences in the elimination rates between subpopulations are substantial enough , i.e. at concentrations where the two PD curves are further apart from each other. This mainly happens closer to the inflection point (midpoint) of the PD curves, where the changes in binding kinetics can create a large enough change in the PD curve for the minority of the population. This is now discussed on page 10 lines 12-17.
The Discussion part seemed to have similar content as the Introduction and do not discuss the results in details. However, the Results part was too lengthy for discussion was made in the Result part. It would be better for the authors to restructure the content in the Results and Discussion part.We have now moved the relevant parts from the results to the discussion (now in page 9 lines 25-33, as well as the paragraph mentioned in point 4 and 5).
The authors should provide a concrete conclusion of the use of this model in practice.We have added a paragraph on the indented use of the approach in practice on page 9 lines 47 to page 10 line 7. There we discuss how the approach can be useful in investigating antibiotic action in vitro where heterogeneities in bacterial populations can be measured (for example, as a result of occasional gene amplification). This would help us e.g. better understand how changes in gene expressions could affect antibiotic susceptibilities
After restructuring, this manuscript can be accepted and would serve as a valuable literature for the pharmacodynamics studies. Although the MIC method may logically have weaknesses, it is very easy and comparable to clinical data. A concrete example or conclusion is needed to see how the model presented in this paper can be used in clinical practice.We agree that currently the MIC method for clinical practice is the most straightforward to use as it has been shown to (weakly) correlate with treatment success. However, as a single variable it has limited predictive power: it cannot explain why treatments could work when they should not be working based on MIC or conversely, why treatments that should work based on the MIC often fail (this has now been added to the discussion on page 8 line 17 to page 9 line 3). This is often attributed to changes in the elimination rates of bacteria over time (for example due to heteroresistance) that is difficult to capture with only the MIC.
Heteroresistance is now frequently discussed as a source of treatment failure and has been recently defined as at least 8 fold MIC difference between bacterial subpopulations (Nature Reviews Microbiology, 2019). Here, our intention was to caution using the definitions of heteroresistance too strictly, as even minor variations in the processes leading to bacterial death can lead to bi- or multiphasic elimination curves even though that does not necessarily cause 8-fold changes in MIC. We have added a paragraph on this in the discussion (page 10 lines 18-27). In addition, we highlight in the abstract (page 1, lines 23-25) as well as in the last sentences of the discussion that our results indicate that heteroresistance is the default scenario rather than a rare biological phenomenon and should therefore always be planned for to achieve sufficient treatment length.
Finally, we have also added an example on how our approach could be used in clinical practice by complementing PK/PD approaches on page 10 lines 1-7.